# A nonlinear relationship between prediction errors and learning rates in human reinforcement-learning

Boluwatife Ikwunne[1], Jolie Parham[1], Erdem Pulcu🔘[1,2,3]*

**1** Psychopharmacology and Emotion Research Lab, Department of Psychiatry, University of Oxford, Oxford, United Kingdom, **2** Computational Psychiatry Lab, Department of Psychiatry, University of Oxford, Oxford, United Kingdom, **3** Neuroscience, Society and Ethics Lab, Department of Psychiatry, University of Oxford, Oxford, United Kingdom

* erdem.pulcu@psych.ox.ac.uk, pulerd@gmail.com

## Abstract

Reinforcement-learning (RL) models have been pivotal to our understanding of how agents perform learning-based adaptions in dynamically changing environments. However, the exact nature of the relationship (e.g., linear, logarithmic etc.) between key components of RL models such as prediction errors (PEs; the difference between the agent's expectation and the actual outcome) and learning rates (a coefficient used by agents to update their beliefs about the environment) has not been studied in detail. Here, across (i) simulations, (ii) reanalyses of readily available datasets and (iii) a novel experiment, we demonstrate that the relationship between PEs and learning rates is (i) nonlinear over the PE/ learning rates space, and (ii) it can be accounted for by an exponential-logarithmic function that can transform the magnitude of PEs instantaneously to learning rates in a novel RL model. In line with the temporal predictions of this model, we show that physiological correlates of learning rates accumulate while learners observe the outcome of their choices and update their beliefs about the environment.

## Author summary

All living agents constantly learn and adapt to changes in their environments, a process normally hidden from observation and often understood through computational models. A key part of this is how we react to "prediction errors" – the difference between what we expect and what actually happens. These differences influence our "learning rate," which is how quickly we update our beliefs about the world, and not much scientific work has been done on the exact relationship between prediction errors and learning rates. Our work demonstrates that this relationship is not always simple, or linear. Instead, we suggest that it is non-linear and depends on different types of uncertainty in the environment. Furthermore,

**Data availability statement:** Model and analysis scripts are available from an Open Science Framework repository: https://osf.io/g7mkv/?view_only=5b4c2ec0eb8442bf95fd-26e151cd71d0.

**Funding:** EP was previously supported by a grant jointly funded by the UK Medical Research Council and Janssen Pharmaceuticals (MR/S035591/1) and currently full-time employed by National Institute of Health Research Oxford Biomedical Research Center. BI is supported by Rhodes Scholarship for West Africa. The pupillometry study was funded by a University of Oxford Department of Psychiatry small research grant awarded to BI and EP. The funders had no role in study design, data collection and analysis, decision to publish, or preparation of the manuscript.

**Competing interests:** The authors have declared that no competing interests exist.

physiological activity measured by recording pupil size during learning suggest that correlations linked to learning rates build up as we observe the outcomes of our actions and adjust our beliefs, supporting our proposed model accounting for how our brains use unexpected events to refine learning.

## Introduction

We are living in a dynamic world in which probabilistic relationships between cues and outcome events create different sources of uncertainty [1] (e.g., not all grey clouds bring rain, and the predictive significance of grey clouds may change from season to season). Living in an uncertain world continuously recruits learning systems in the brain, with which agents try to make better decisions in their environments. Behavioural adaptation to dynamically changing environments is a type of value-based decision-making which is critical for survival in the wild, and long-term evolutionary fitness. Consequently, reinforcement-learning (RL) models describing cognitive/computational processes underlying learning-based adaptations have been pivotal in behavioural [2–4] and neural sciences [5–10], as well as machine learning [11,12].

In cognitive neuroscience of RL, a wide majority of the literature focuses on neural/physiological correlates of prediction errors, whereas the relationship between prediction errors and learning rates, and generally the neural/physiological correlates of learning rates remains mostly understudied [13]. In terms of operational definitions, learning rates determine the speed with which agents update their expectations about the environment based on prediction errors they experienced. It is possible that the reason why the relationship between prediction errors and learning rates is understudied may be a byproduct of a lack of approachable models which would allow investigation of the relationship between prediction errors and learning rates, instead of a lack of interest in this relationship/research question. In this paper, we describe two simple RL algorithms that can adequately address this knowledge gap. We describe the properties [14] of these simple update rules, and test their applicability to different experimental tasks. The main motivation of our work is to inform a wider discussion in the field of RL towards development of more usable, relatable and intuitive computational models that can make investigations into the relationship between learning rates and prediction errors accessible to scientists across cognitive neuroscience, experimental psychology, and computational biology and psychiatry.

Let us start by considering the factors which influence learning rates, as evidenced by literature that utilised existing RL algorithms. A number of previous studies showed that human learners adjust their learning rates based on the information content of outcomes [15,16], utilising a higher learning rate for outcomes more informative for predicting the future state of the environment. In computational neuroscience, learning rates are commonly modelled by using different RL algorithms [13,17,18] —which are in effect approximations of the Bayesian models [19,20] that can also estimate

otherwise hidden generative rules of dynamically changing environments. The Rescorla-Wagner (RW) algorithm is undisputedly the most widely used RL model [17] in the literature:

$$p_{(t+1)} = p_{(t)} + \eta(O_{(t)} - p_{(t)})$$

(1)

where $p$ is the agent's probability estimate for an outcome, letter $O$ designates the actual outcome (e.g., win vs null), $\eta$ is the learning rate and $t$ is the trial number. However, a key limitation of the RW model is that it is commonly fitted to participant choices within a block of an experimental task, to be able to estimate participants' average learning rates during that period. This analytic approach uses the RW model as an assessment tool to measure the speed of learning, implemented by a number of key RL studies [5,15,16,19] consistently showing that human learning rates are higher in volatile (i.e., environments in which contingencies change), relative to stable environments in which contingencies stay the same, or do not change very frequently. Similarly, learning rates also change [nonlinearly] as a function of whether PEs are positive or negative [21,22]. Globally, these findings clearly indicate that learning rates are not a constant coefficient. As a result, approximating a single learning rate from a block of an experiment may not be the most ideal approach for understanding mechanisms underlying learning-based behavioural adaptations. The family of hybrid Pearce-Hall models [13,18,23] aim to mitigate this limitation by updating learning rates on a trial-by-trial basis. For example, one common implementation is based on a weighted integration of the learning rates from previous trials with the magnitude of the absolute value of the prediction errors (PEs, i.e., the difference between the agent's expectation and the real outcome):

$$\eta_{(t+1)} = \omega\eta_{(t)} + (1-\omega)\left|O_{(t)} - p_{(t)}\right|$$

(2)

where $\omega$ is a free parameter estimated between 0 and 1 and determines how much weight is given to the learning rate ($\eta$) on the current trial and the absolute value of the PE (i.e., the unsigned PE) to update the learning rate into the next trial. Here, it is important to highlight that in the majority of the RL models, the absolute value of the PEs ($\left|O_{(t)} - p_{(t)}\right|$) is commonly used as a proxy for the degree of surprise associated with observing the outcome in question. The model would then update the agent's estimates of the reward probability into the next trial:

$$p_{(t+1)} = p_{(t)} + \gamma\eta_{(t)}(O_{(t)} - p_{(t)})$$

(3)

Where $\gamma$ is another free-parameter commonly used for smoothing the effect of the trialwise learning rates which are updated by the surprise signal, therefore finetuning the agent's estimation of outcome/reward probability. While both RW and Pearce-Hall models are commonly used in the literature, these were not developed ground-up considering how the relationship between prediction errors and learning rates should be (e.g., linear, logarithmic or exponential, etc.).

### Learning-based behavioural adaptations in response to different sources of uncertainty

Existing literature suggests that different sources of uncertainty dictate the speed of learning which would be optimal in a given environment [1,24,25]. *Expected uncertainty* relates to intrinsic variability in the occurrence or the nature of an event over and above what could be explained by any learned association. In the literature expected uncertainty is also referred to as aleatoric uncertainty [26], environmental noise [25] or stochasticity [27]. For example, if an outcome follows a stimulus on only 30% of occasions, or if the magnitude of the outcome is inherently variable (e.g., day-to-day changes in air temperature) [1], it would be difficult to accurately predict the frequency or the magnitude of the outcome on any given occasion. In other words, the higher the expected uncertainty the less informative each particular event is; since expected uncertainty erodes the information content of individual events [1,28]. Surprise signals also decay in environments with high expected uncertainty, as prediction errors arising from violation of one's expectations become less informative. In environments dominated solely by expected uncertainty, the optimal learning behaviour would be to estimate cue-outcome

associations over a long period of time utilising a relatively low learning rate that reflects the limited epistemic utility of observations for fine-tuning predictions. However, there can be instances when learnt associations are rendered out-of-date relative to the underlying relationships that currently prevail in the environment, since these relationships can change systematically over time, or change suddenly for a shorter period of time. In these cases, the true association between cues and outcomes is more difficult to estimate because it is not stable. This is commonly known as *unexpected uncertainty* (or "*volatility*" [19]). Unexpected uncertainty effectively reduces how informative the previous events are during learning. In other words, previous events become increasingly less informative the higher the unexpected uncertainty in the environment. A real-life example could be a sudden increase in UK daily temperatures way above the seasonal average, brought along by an Atlantic storm. The optimal behavioural adaptation to this type of uncertainty is to put more weight on recent events, as events further back in history are likely to be less informative about current or future state of the environment. Particularly in environments with high unexpected and low expected uncertainty, tuning learning rates to the prediction errors (PEs) would be an efficient way of interacting with the environment. This is because during periods of low expected uncertainty (i.e., if the environment is not noisy), the PEs would be relatively small, and agents can discount them by utilising a lower learning rate. However, when the PEs are reasonably high, this would signal that there must be a dramatic change happening in the underlying structure of the environment. In these situations, agents can quickly adapt by utilising a higher learning rate. Although these conceptual descriptions indicate a relationship between PEs and learning rates, the exact nature of this relationship is yet to be explored in detail.

Historically, learning under expected and unexpected uncertainty has also been investigated by Bayesian models [19,20,27], including our own [25,29], and their approximations [30,31]. These generative models are particularly useful for delineating the true source of environmental uncertainty (i.e., whether it is attributable more to expected or unexpected changes). Although these models are important to mention, they are often complex (i.e., despite some of them being openly available online, they can be regarded as "black box" algorithms for the wide majority of users) and not necessarily plausible representations of computations underlying human learning. For example, emerging evidence from our lab suggests that, at best, human learners behave similarly to a degraded/lesioned Bayesian learner [32]. Furthermore, to the best of our knowledge, there is no consensus in the literature as to how these models should be fitted to human choice behaviour. For example, in their seminal work Behrens and colleagues [from whom we obtained original data and re-analysed below] used a Bayesian Ideal Observer model to create regressors that track environmental volatility for their neuroimaging analysis, but did not fit this model to participant choice behaviour as the latter would require making many [arbitrary] assumptions about whether these models should have similar risk or probability weighting preferences as human participants or whether they should also act optimally in these behavioural domains, or how to perform model comparison considering these models do not necessarily have free parameter in the traditional sense, etc. Consequently, it is clearly beyond the scope of the current work to reconcile the differences between these complex models and to come up with a single unifying model that can explain human choice behaviour the best. Here, instead, we focus on simple and relatable learning algorithms that can perform a dynamic learning rate update based on a nonlinear relationship between learning rates and the magnitude of the PEs. We think the models that we explore in the subsequent sections can empower cognitive scientists, psychologists and clinicians with simple, relatively easier-to-implement tools to allow investigation of questions surrounding learning rates during RL.

## Results

### A parabolic relationship between learning rates and prediction errors

In RL tasks with probabilistic contingencies and binary outcomes (i.e., reward versus no reward), the PEs are always between -1 and 1 (note that this formulation only applies to tasks with binary outcomes, the range of PEs would otherwise be determined by the reward range in the environment). For example, if one option available in the environment returns no rewards (i.e., a vector of null outcomes: [0, 0, 0, 0...]) yet the agent continues to bet on that option thinking

their luck will turn, the lowest values possible for the PEs experienced during that period can be is a vector of -1s (i.e., if the agent remains implacably confident throughout that the outcome surely must occur on this next occasion). Another widely established assumption is that there should be no additional learning when the PE is zero. That is if the agent can consistently predict outcome events with 100% accuracy, then there is nothing new to be learned from the environment, and the effect of a constant learning rate should diminish (Eq 1). In environments with high unexpected but low expected uncertainty, one would assume a somewhat linear relationship between prediction errors and learning rates (in a V shape designated by the black dashed lines in Fig 1), meaning that agents should update their beliefs proportionately to how far away their beliefs about the environment were relative to the reality. However, also considering the nonlinearity in wide majority of cognitive processes (e.g., temporal discounting [33,34], risk perception [35,36], stochastic choice [37,38] etc.) across human and non-human primates, a simple functional form satisfying all of these assumptions proposes a parabolic relationship between the PEs and learning rates:

$$\eta = \left(O_{(t)} - p_{(t)}\right)^2 / 4\kappa$$

(4)

In this notation 4 in the denominator ensures that the focal point of the parabolic curve on the $xy$ plane is always (0, κ). The focal point of the parabola adjusts how steeply PEs transform to learning rates (Fig 1), and it is the free parameter to be estimated from participant choices.

Then, by combining Eqs 1 and 4, the update algorithm (i.e., updating PEs with corresponding learning rates on each trial) can be expressed as:

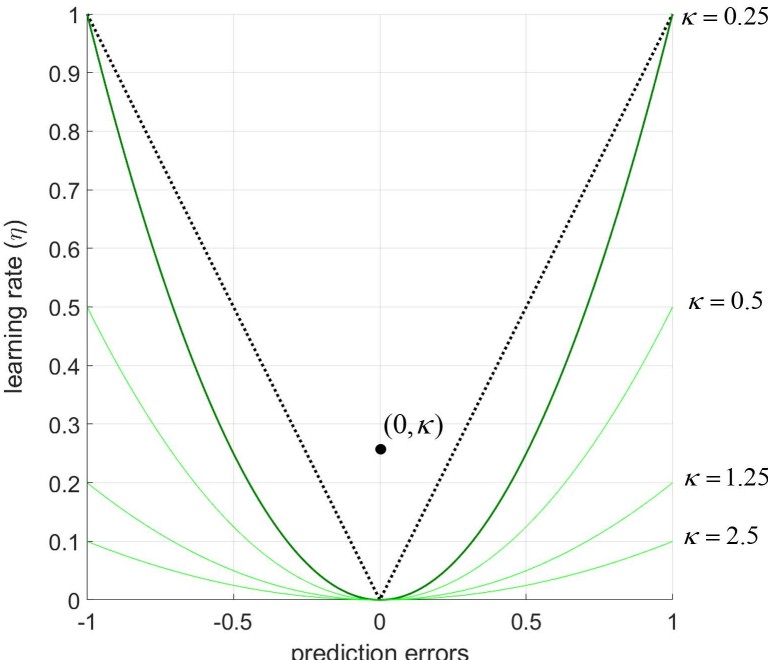

**Fig 1. A parabolic relationship between prediction errors and learning rates obtained by a cubic learning algorithm.** The proposed algorithm approximates this relationship under the control of a single free-parameter (thick green line, κ = 0.25). The free-parameter κ also determines the focal point of the parabolic curve (i.e., the black dot at (0, 0.25) on the $xy$ plane is the focal point for the thick green parabolic curve). Increasing values of κ flattening the relationship between prediction errors and learning rates, leading to lower learning rates for a given magnitude of the prediction error. A few examples are illustrated by light green lines at κ = 0.5; 1.25 and 2.5. Parameter values smaller than 0.25 could allow prediction errors to be converted to learning rates even more steeply.

$$p_{(t+1)} = p_{(t)} + (O_{(t)} - p_{(t)})^3 / 4\kappa \qquad (5)$$

We will refer to this model as the cubic model in the rest of the manuscript. Here, the free-parameter κ cognitively accounts for individual variability in how steeply learning rates increase at a given magnitude of PE. This cubic update rule would allow agents to adjust their behaviour dynamically as soon as they encounter any unexpected environmental changes (i.e., those leading to higher PEs). However, in noisy environments (i.e., those with high expected uncertainty) scaling learning rates to the PEs would not be an ideal way to learn. This is because, in stable environments with high expected uncertainty, agents need to discount PEs as these would be noisy and not informative about the future state of the environment. The model can also account for this behaviour at higher values of κ (Fig 1). From this perspective, the parameter κ is also under the influence of the agent's global inference about the level of expected uncertainty (i.e., noise or the standard deviation (σ) of the generative process) characterising the environment [24]. A recent analysis of normative learning models demonstrated that learning rates decay exponentially in noisy environments (i.e., those with high expected uncertainty) [39], as expected uncertainty erodes the information content of outcomes, and therefore impair the relationship between PEs and learning rates. In these situations, the agents should no longer attribute high magnitude prediction errors as indicating informative changes in the environment, but should instead simply treat them as noisy fluctuations which are not informative. Note that in noisy environments in which the rewards are delivered in the normalised space (i.e., between 0 and 1), maximum entropy is reached if the mean reward rate is 0.5±σ. Therefore, it is also plausible to assume that the higher-level relationship between expected uncertainty (i.e., environmental noise (σ)) and the free-parameter κ can be in an exponential form:

$$\kappa = f(\sigma) = ae^{\beta\sigma} + \varepsilon \qquad (6)$$

which would allow the model to tune learning rates with respect to the magnitude of prediction errors at a given level of environmental noise. Note that in this formulation $a$ designates the initial level of expected uncertainty in the environment ($a \neq 0$). Under these assumptions, the nonlinearity of learning rates guiding behavioural adaptations could resemble Fig 2, below. The distribution of these simulated learning rates is globally in line with empirical results reported by other research groups which showed a linear-stochastic relationship between learning rates estimated from participant choices and prediction errors as a function of expected uncertainty/environmental noise [30].

### A data driven approach for evaluating the nature of nonlinearity between learning rates and prediction errors

Another approach with which one can extract trial-wise learning rates is by exploiting the mathematical properties of a 2-parameter [probability] weighting function [14]. This approach would assume an exponential-logarithmic relationship between the unsigned PEs and learning rates and allows even more flexibility than the parabolic relationship described by the cubic model above, as it is free of any *a priori* assumptions about the exact shape of the relationship between PEs and learning rates:

$$\eta = e^{\left(-\delta\left(-\log\left(|O_{(t)} - p_{(t)}|\right)\right)^{\lambda}\right)} \qquad (7)$$

and the rest of the update rule remains identical to the RW model (Eq 1). The parameters δ and λ are to be estimated from participant choices (both parameters can be estimated within boundaries [0, ∞], Fig 3). These parameters work in tandem to reveal the trajectory of the weighting curve. This model, which can cover the whole PE-learning rate space with different nonlinear trajectories (Fig 3), can help with further scrutinising whether the relationship between PEs and learning rates is indeed parabolic (Eq. 5), or linear as inherently proposed by the Rescorla-Wagner model. This model describes that agents will tune their learning rates in response to different PE magnitudes. The flexibility of

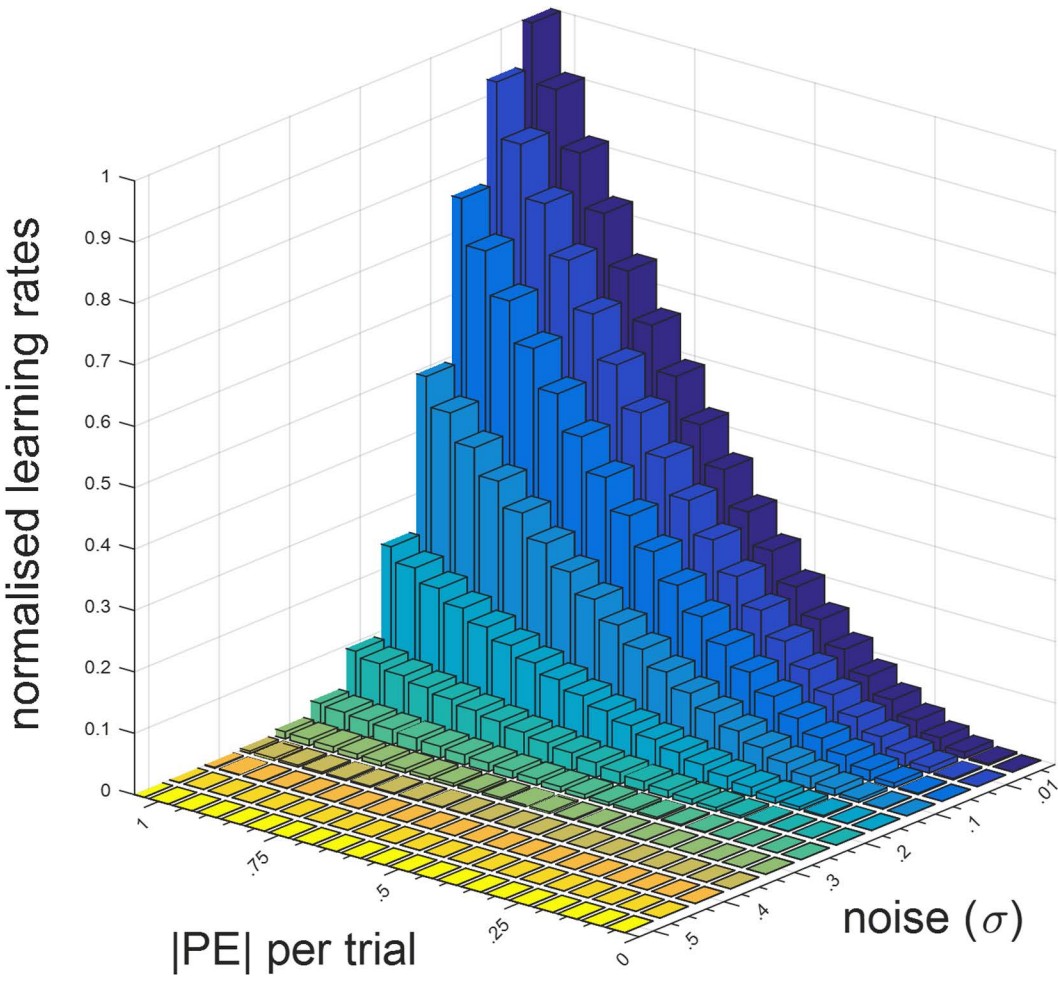

**Fig 2. Nonlinearity in trial-wise learning rates as a function of prediction errors and environmental noise.** One viable function, perhaps out of many possible solutions, accounting for the relationship between κ and σ is $\kappa = f(\sigma) = \frac{\pi e^{7\pi\sigma}}{10^3} + \frac{\gamma}{6}$. This would suggest that the parameter K is a function of environmental noise, and one example of this exponential relationship is shown in **Fig A in** S1 Text. The figure demonstrates that the σ parameter can be estimated reliably between .01 and .4, covering the whole learning rate space. Note that for outcome magnitudes within the normalised space (i.e., [0 1]), values of σ higher than .3 would generate outcomes outside these boundaries. Here, ϒ denotes the Euler-Mascheroni constant.

this non-linear model allows learning rates to slow down or ramp up in response to environmental demands as it can take many different trajectories. It is worthwhile to highlight that cognitively this model serves a similar function in tuning learning rates to the magnitude of the PEs that we described above for the cubic model. Therefore, it is possible to conceptualise this model as a more flexible variant of the cubic model without any *a priory* [parabolic] assumption. Furthermore, the model can also fully reduce to the traditional RW model, a quantitative feature which is not available in the cubic model, as the model is able to generate constant learning rates over the prediction errors, as shown by horizontal lines in Fig 3A. In our own evaluation, these features make the exponential-logarithmic model better than the cubic model.

In the subsequent sections, we simulate the behaviour of these [cubic and exponential-logarithmic] models to evidence how they can flexibly adapt to reward outcomes in dynamically changing environments, followed by demonstrating their performance in RL tasks with binary and continuous/gradient outcomes.

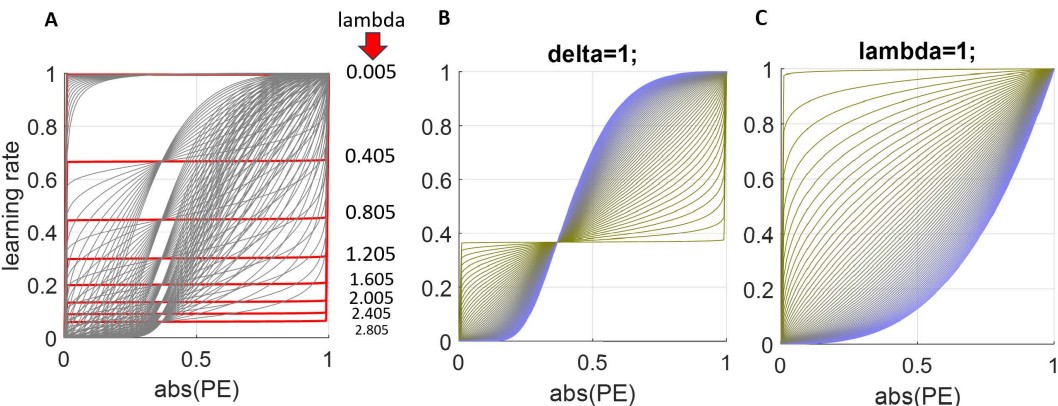

**Fig 3. Nonlinearity in trial-wise learning rates as a function of prediction errors under an exponential-logarithmic assumption. (A)** The parameters δ and λ from Eq 7 are coarsely sampled for illustration purposes. When the parameter combinations produce horizontal lines over the PEs (i.e., the x-axis), the model is fully reduced to the Rescorla-Wagner rule with a constant learning rate (Red thicker lines, lambda = 0.005, variable delta values shown in figure). **(B-C)** Change in learning rate vs absolute value of prediction error (PE) trajectories when the value of one of the parameters in the model is fixed to 1. Although it is important to highlight that these parameters interact with each other to set the trajectory of the learning rate/absolute value of prediction error relationship, higher the lambda value the more sigmoidal this relationship will be as opposed to a constant learning rate, whereas higher the delta, more parabolic will be the relationship between prediction errors and learning rates.

**Simulated data.** First, we aimed at testing the upper boundary of how well simulated agents could perform in dynamically changing environments under different learning models. This could be understood as a metric of model flexibility, which is an important aspect of model identification that can describe adaptive learning behaviours. We wanted to test whether the novel models that we proposed can behave at least on-par with traditional models like the RW model. We simulated the behaviour of RL models (i.e., Rescorla-Wagner, hybrid Pearce-Hall, cubic and exponential-logarithmic models) in a task environment in which unexpected (i.e., volatility) and expected (i.e., noise) uncertainties were manipulated independently in different segments of the task environment (2x2 factorial design: high volatility low noise, low volatility high noise covering all possibilities, Fig 4). On each trial, the task environment returns a reward outcome between 0 and 1 (Fig 4, y-axis). The outcomes were generated to be variable from one trial to the next. The first 300 trials are designed to be volatile (i.e., the mean of the underlying Gaussian distribution jumps to a different point in every 100 trials) with low noise (SD = 0.05). The subsequent 200 trials are a low volatility environment (i.e., the mean of the Gaussian distribution generating the rewards stay the same) with a higher level of noise (SD = 0.1). After 500 trials, the environment becomes volatile again, this time with high noise level (SD = 0.10). The next 200 trials are a low volatility environment with low noise (SD = 0.05). In the last 500 trials the environment is noisier (SD >= .15), which further obscures the transition between stable and volatile periods. Analytically, the learning rates were estimated under each RL model by minimising the average PE per trial throughout the simulated task environment (from all 1500 trials all at once). In our initial analysis, we performed a procedure for minimising prediction errors based on the full trajectory of outcomes (including the future state of the environment) rather than adapting learning rates based on local/past information, as at this stage we did not model agents making decisions under different models and arbitrarily assigned learning rates. Our approach is akin to a maximum likelihood estimation from a search of viable discrete parameter values, as minimising the PEs under a given RL model indicates how well the model-agents' beliefs about the reward rate can converge to the mean of the actual outcome distribution. In these simulations the effect of choice stochasticity was not considered as choices were not simulated with arbitrarily selected learning rate values (which would in effect be a theoretical equivalent of a generate-recover simulation).

In the simulated task environment, all models converge reasonably well with the true average reward rate (different coloured lines designating different models globally overlap in Fig 4). In line with this qualitative/visual evaluation, the

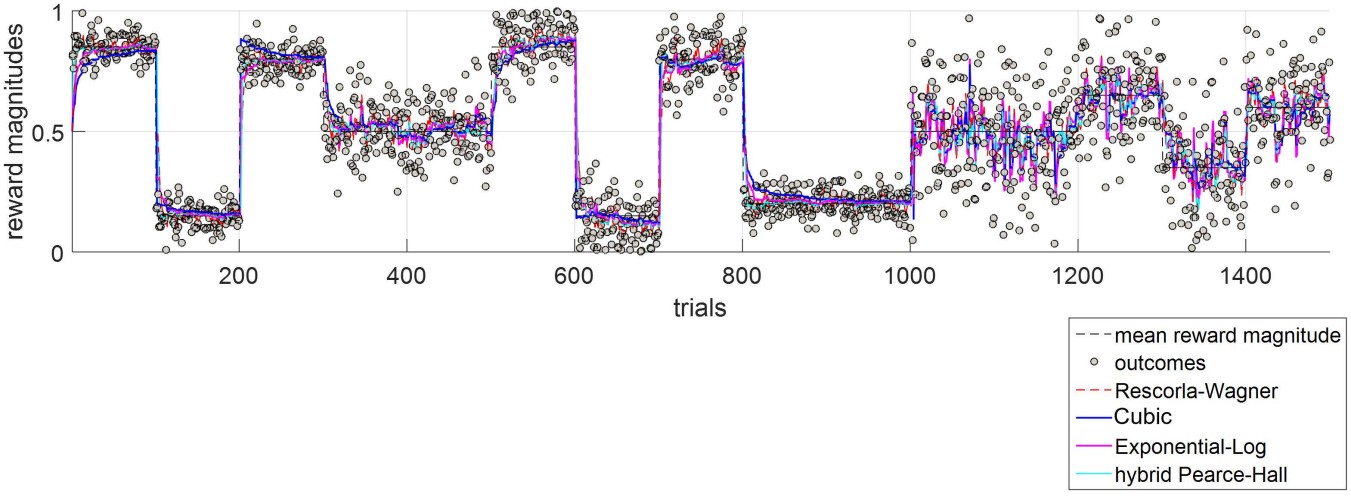

**Fig 4. The behaviour of different reinforcement-learning models in a task environment in which unexpected and expected uncertainties were independently manipulated.** All models converge reasonably well with the actual mean of variable rewards. The learning rate for the Rescorla-Wagner model (η, Eq 1) is 0.32. For the hybrid Pearce-Hall model, ω (Eq 2) is 0.48 and λ is (Eq 3) is 1.56. For the cubic model κ is 0.11 (Eqs 4-5). For the exponential-logarithmic model, the parameters δ and λ are 0.83 and 1.45, respectively (Eq 7). Because models perform ever so comparably, their differences are illustrated in **Fig B in** S1 Text, showing the average prediction error values relative to the simulated outcomes in the task environment. Note that, the simulation environment shown was generated only once, covering many possibilities of environmental volatility and noise, and their interaction, whereas the models were fitted iteratively until parameters minimising the average magnitude of the prediction error relative to the actual outcome sequence are identified.

magnitudes of prediction errors under each model were not statistically different (F (4, 7495) =0.32, p = 0.863, Fig B in S1 Text), meaning that all models can perform behavioural adjustments comparably well. The nonlinear relationship between PEs and learning rates under the cubic, the exponential-logarithmic and the hybrid Pearce-Hall models are illustrated in Fig C in S1 Text.

The hybrid Pearce-Hall, the cubic and the exponential-logarithmic models can estimate learning rates on trial-by-trial basis. The PEs and learning rates estimated under each model (based on model behaviours shown in Fig 4) were decorrelated (-.01 < r (1498) <.01, all p >.80). However, learning rates estimated under the hybrid Pearce-Hall model (mean±SD = 0.096 ± 0.071) were significantly higher than the ones estimated under the cubic (mean±SD = 0.042 ± 0.094) and exponential-logarithmic models (mean±SD = 0.076 ± 0.12, F (2, 4497) = 119.07, p <.001). Nevertheless, these learning rate estimates from different models were very significantly correlated (all r (1498)>.68, p <.0001). Taken together, these results indicate that 1-parameter cubic and 2-parameter exponential-logarithmic models can be viable alternatives to the existing RL models. Here, it is worthwhile to highlight that one of the main differences between these models is how quickly they would adjust their behaviours in response to environmental change. Under the cubic and exponential-logarithmic models, learning rates accelerate as soon as there is a dramatic environmental change, therefore leading to large PEs, and learning rates subsequently decay as the magnitude of PEs get smaller. Comparatively, under the hybrid Pearce-Hall model acceleration and deceleration of learning rates happen relatively more gradually. This is because learning rates here are weighted integration of learning rates and the PEs from the previous trials (Eq. 3, Fig D in S1 Text), as determined by a weighting constant (Eq 2) that is estimated from participant choices. Although each of these three models (Fig C in S1 Text) describe an efficient way of interacting with the environment (i.e., accelerating learning rates only in cases in which the changes in the environment are informative), notably, they make different predictions about the temporal properties of learning rates (Fig D in S1 Text). In the hybrid Pearce-Hall model the information is fed forward as the learning rate is a secondary process, one which is updated in parallel (Eqs 2–3) in addition to estimated reward rate.

In contrast, the cubic and exponential-logarithmic models both assume that incoming prediction errors would trigger their corresponding learning rates instantaneously during the feedback period of the current trial. We tested these divergent predictions on temporal properties of learning rates subsequently in a novel RL experiment with pupillometry.

**Human behaviour.** First, we investigated how well nonlinear RL models explain human behaviour in two previously published data sets. The first dataset contained healthy volunteers (N = 18) who completed a RL task with stable (120 trials) or volatile (170 trials) reward contingencies in different task blocks [19]. In the original work by Behrens and colleagues [19], participants were asked to choose between two fractals which returned variable reward amounts on each trial or a null outcome, based on the underlying reward probabilities that participants needed to learn. In the learning task, the reward magnitudes associated with each fractal were generated randomly. In the current reanalysis of this data, the stochastic choice/sigmoid function fitted to participant choice behaviour was identical across all RL models. More specifically, participants' decision values were computed by integrating their reward probability estimates (computed under each RL model) with their corresponding reward magnitudes associated with each option. The reward magnitudes (m) associated with each shape were modulated by a single power utility parameter ($\rho$) accounting for participants' risk attitude [35], such that $m^? = m^\rho$. Then, the expected value difference between two options was calculated and fed into a sigmoid function to generate the choice probabilities [37]. This is a very common approach in modelling participant choice behaviour in this task.

Overall, all RL models estimated higher learning rates in the volatile relative to the stable block, replicating the findings previously reported by Behrens and colleagues (2007). Analysis of model log likelihoods by fitting a 2x4 repeated measures analysis of variance (ANOVA) did not reveal any significant main effect of block type (i.e., volatile or stable; $F_{(1,68)}=.014$, $p=.907$) or model ($F_{(1,68)}=.018$, $p=.997$), or any significant block type x model interaction ($F_{(3,68)}=.002$, $p>.999$). Pairwise comparisons further confirmed that the models were all statistically comparable (log likelihood values are only differentiated at the 3rd decimal place, $p>.923$), suggesting that all models tested here fit both volatile and stable environments comparably well. Model comparison metrics which penalise for number of parameters such as Akaike's Information Criterion favored the hybrid Pearce-Hall model (AIC = 84.486) over Rescorla-Wagner model (AIC = 84.888) and cubic model (AIC = 84.931), and finally the exp-log model (AIC = 85.726). Due to such small margins between the models, Bayes Omnibus risk calculated for these comparisons is 0.943 indicating that it is difficult to discriminate between model fits.

The difference in learning rates between stable and volatile blocks were only statistically significant under the RW and cubic models ($t(34)$ values 2.114 and 2.079, p values 0.042 and 0.045, respectively). The average learning rates from the hybrid Pearce-Hall and the exponential-logarithmic models did not capture a statistically significant volatility effect ($t(34)$ =1.057 and 1.460, $p=.298$ and.153, respectively). Similar to the simulated results reported in the preceding section, the hybrid Pearce-Hall model generated much higher learning rate estimates relative to other models (Fig 5A). The analysis with the cubic model indicated that changes in environmental volatility did not affect the relationship between PEs and learning rates (Fig 5B).

Despite being a frequently used analytic approach in the literature, one might argue that block-wise fitting of these models would create a mismatch between the learning strategy employed by the participants versus the models, such that models would be informed about the structure of the task ahead of time. Because, to the best of our knowledge, the blocks were administered continuously by Behrens and colleagues, a continuous fitting approach would ensure that learning models are also agnostic to the information about the task structure. Furthermore, the block-wise fitting approach implemented in the preceding section may also work against the strengths of the learning models that can perform dynamic learning rate update. We predicted that a continuous fitting approach would compromise the RW model to a greater extent, as it cannot perform a dynamic learning rate update to adapt to changing outcome schedules between stable and volatile blocks. However, this was not supported by strong evidence, as models remained highly comparable based on log likelihood values ($F(3, 71)=.012$, $p=.998$). Other model goodness of fit measures which penalise for

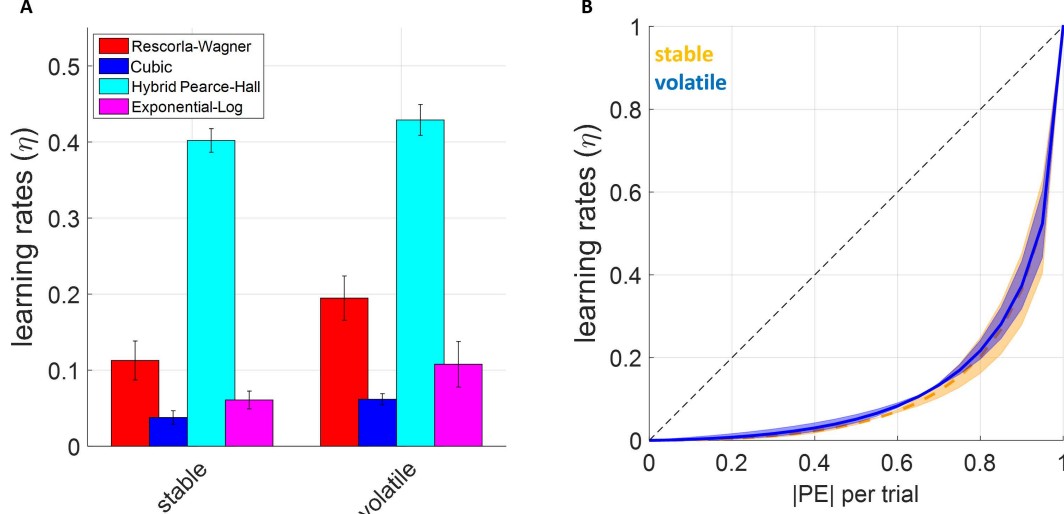

**Fig 5. The learning rates estimated under each reinforcement-learning model. (A)** Hybrid Pearce-Hall model estimates higher learning rates relative to the Rescorla-Wagner model, whereas cubic and exponential-logarithmic models estimate overall lower learning rates. **(B)** The relationship between PEs and learning rates estimated by the exponential-logarithmic model reveals a nonlinear trajectory very similar to what is proposed by the cubic model. Volatility of different task blocks did not seem to influence learning rate trajectories in terms of how absolute values of the PEs tended to influence learning rates. This relationship was similarly exponential in form across both stable and volatile task-blocks.

number of parameters such as sum of Akaike's Information Criterion (AIC) favoured the cubic model which had in total 3 free parameters (i.e., the parameter κ which determines trial-by-trial learning rates, power utility, inverse temperature), by a margin of 0.31 units to the runner-up model which was hybrid Pearce-Hall which had in total 4 free parameters. Bayes Omnibus risk calculated for these comparisons remained identical to fitting the models individually to each block, as stated above.

Using a continuous fit approach, it is also possible to [re-]investigate the volatility effect on human learning rates by taking the summary/mean value of trial-wise learning rates that fall within stable versus volatile blocks (note that this approach only applies to models that can generate trial-wise learning rates, namely cubic, exponential-logarithmic and hybrid Pearce-Hall models). This approach did not identify any significant volatility effect ($F_{(1, 51)} = .01$, $p = .919$) or volatility by model interaction ($F_{(2, 51)} = .172$, $p = .842$), but a significant main effect of model influencing the exact parameter estimates (($F_{(2, 51)} = 185.56$, $p < .001$). As previously shown in Fig 5A, the estimates of the Pearce-Hall model were substantially higher than those of the other models. It is possible that the lack of volatility effect using this approach may also be due to an imbalance of trial numbers in the original design of Behrens et al, i.e., 120 trials in the stable block versus 170 trials in the volatile block.

Taken together, these results suggest that the models that we newly proposed behave reasonably comparably to the existing RL models such as RW and hybrid Pearce-Hall, and in the context of learning probabilistic reward associations that deliver variable reward amounts their performance cannot be reliably differentiated.

## Exponential-logarithmic model provides greater flexibility capturing behavioural adaptations

In order to tease apart the differences between models that can estimate trial-wise learning rates further, we analysed another set of previously published data [40]. In this experiment, Vaghi and colleagues (2017) reported results from 25 healthy volunteers who tried to catch a particle initiating from the centre of the circular grid by correctly positioning a virtual bucket in order to collect as much points as possible (i.e.,"the bucket task", Fig 6A). Participants try to catch blue particles

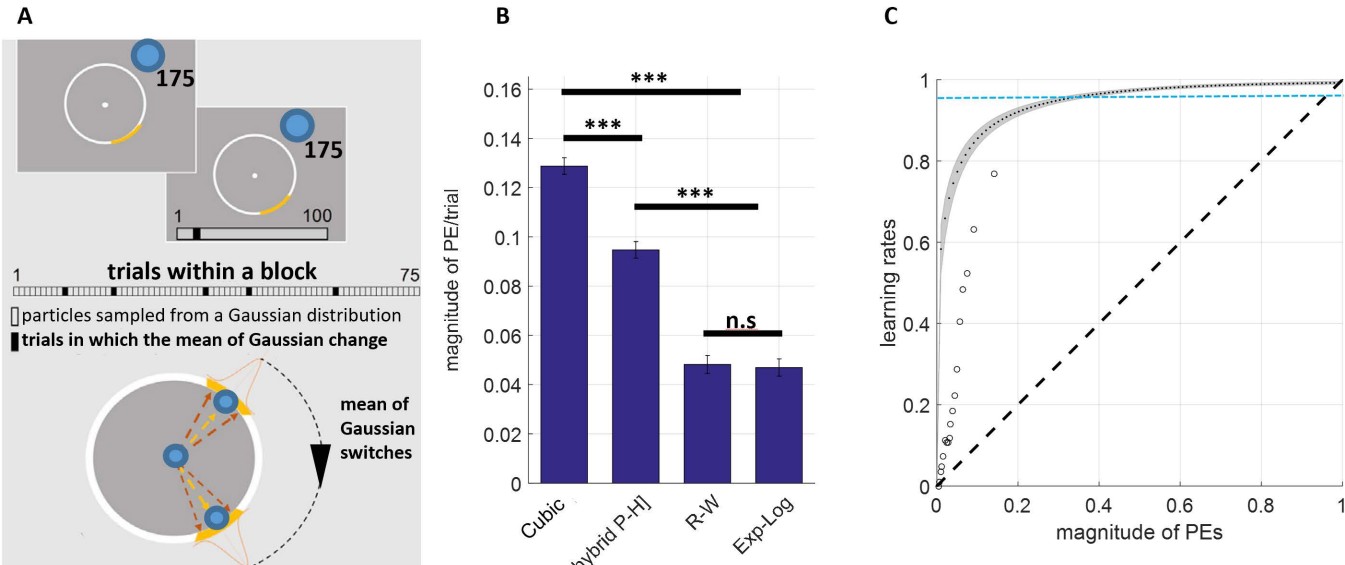

**Fig 6. Learning rate results from the reanalysis of the bucket task. (A)** Overview of the bucket task. **(B)** The analysis of the data with the novel reinforcement-learning models demonstrated that, among models that can estimate trialwise learning rates, the exponential-logarithmic model exhibits greater flexibility to account for participant behaviour (i.e., the model minimising the deviation from participants bucket placement) relative to the cubic model which assumes a strictly parabolic relationship. The values are normalised from 360° to [0,1].***p<.001 **(C)** The model estimates steep learning rate increases over the prediction error space and results mostly align with those reported by Vaghi et al., (2017). Average model-free learning rates reported by Vaghi and colleagues indicated by circular markers, curve with grey shading denotes the trajectory estimated by the exponential-logarithmic model. Here, population average of learning rates estimated by the RW model is 0.968±0.034 (mean±SD, mean shown by the blue horizontal dashed line). Consequently, the exponential-logarithmic model offers greater flexibility at the lower PE values covering the spectrum where most of the actual PEs in this task were, whereas learning rates from these competing models somewhat converge at the higher PE values (i.e., top right of panel **C**).

initiating from the centre of a circular grid with a virtual yellow bucket in order to bank as much points as possible. The particle lending trajectories are drawn from a Gaussian distribution with a mean that jumps to a different point randomly (i.e., the switch points). Participants need to adjust their inferences in order to place their buckets to where the particles are most likely to land. This experimental design is somewhat similar to the simulated experiment shown in Fig 4. Participants completed 300 trials in total across 4 blocks of 75 trials, which meant that overall the environment changed more frequently than our simulated environment. Although the trajectory of the particles originating from the centre of the circular grid changed to a new point randomly, the standard deviation of the landing points remained stable (i.e., no additional experimental manipulation of expected uncertainty/environmental noise).

In this scenario, we show that the exponential-logarithmic model can account for participant behaviour significantly better than the cubic model (t(48) = 14.168, p<.001) and the hybrid Pearce-Hall model (t(48)= 9.514, p<.001). Although comparisons based on average PE magnitude illustrates that the exponential logarithmic model behaves comparably with the Rescorla-Wagner model (p=.807, Fig 6B), metrics such as AIC that penalise the number of parameters would still favour the exponential logarithmic model ever so slightly (group-wise sum of AIC values 2260 vs 2260.2). However, as highlighted before, the R-W model would not allow investigation of the relationship between learning rates and prediction errors. The reason why exponential-logarithmic model performs better than the cubic model becomes evident when we plot the trajectory of the estimated learning rates over the range of the prediction errors— participants utilise increasingly higher learning rates which gradually plateau when the magnitude of prediction errors exceeds 0.2, also revealing a non-parabolic trajectory. In tasks with more precise outcome feedback such as the bucket task, the learning rate trajectory is remarkably different relative to traditional reinforcement-learning experiments in which outcomes are binary (i.e., win

versus no-win). These results are widely in agreement with the initial model-free (i.e., how much participants updated their bucket positions from the previous trial) learning rates reported by Vaghi and colleagues (2017, circular data points shown in Fig 6C).

## Physiological correlates and temporal properties of learning rates

We also conducted our own learning experiment in an independent cohort to investigate the physiological correlates and temporal properties of learning rates versus prediction errors in a task environment with volatility and variable degrees of noise. Thirty-two healthy volunteers (see Table A in S1 Text for demographic features) completed a reward magnitude learning task which was adapted and simplified from an affective learning task that served as a control experiment reported in one of our earlier studies [16]. On each trial, participants were asked to choose between two abstract shapes and obtained rewards. The reward associated with the chosen shape was represented as a green filling inside a bar which designated 100 available points per trial. This maximum reward amount was distributed between the options, such that if one shape had more reward points (green filling), the other one would have less (i.e., empty grey area within the bar, Fig 7). Participants completed 4 blocks of 60 trials each, covering a 2 x 2 factorial design of high/low volatility and high/low noise, making the task environment similar to the simulation environment shown in Fig 4. Blocks were separated by a brief resting period and participants were informed that the outcome schedules might change in each block. Each block also contained novel shape pairs, visually distinct from the shapes that participants already learnt about. In this experiment, because the participants took breaks between distinct task blocks, fitting the RW model separately to each task block can be deemed less problematic.

Fitting the RW model separately to each task block to assess learning rate adaptation to volatility and noise suggested that participants behaved near-optimally (Fig Fi in S1 Text, also see Pulcu & Browning, 2025 for optimal learning rate adjustment performed by a Bayesian Ideal Observer model in task environments where volatility and noise are independently manipulated [32]). A formal statistical analysis using a repeated measures ANOVA indicated a significant main effect of volatility ($F_{(1, 29)} = 9.484$, $p = .005$) and a significant interaction between volatility and noise ($F_{(1, 29)} = 5.777$,

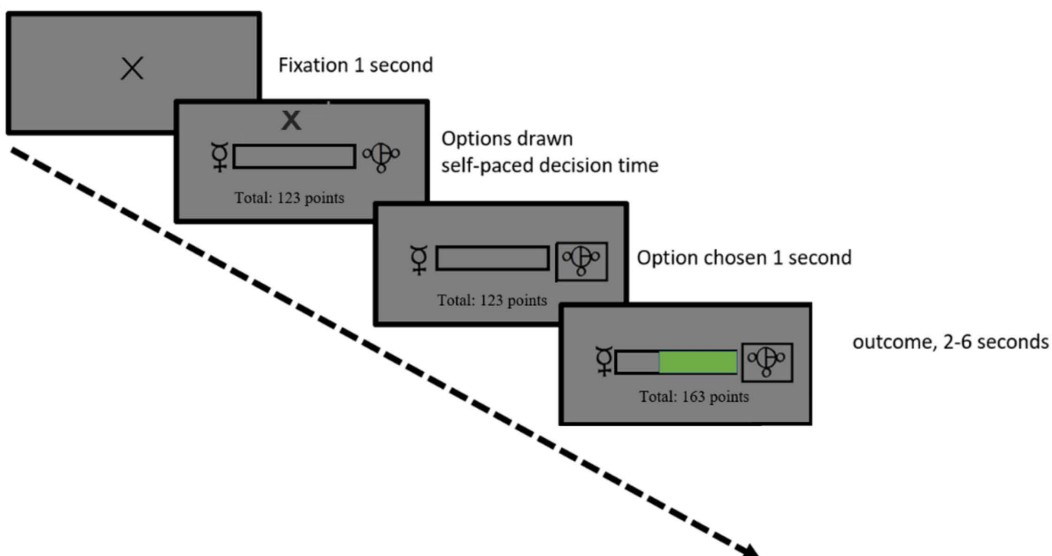

**Fig 7. The timeline of the reward magnitude learning task.** See **Fig E in** S1 Text for the task environment, more details about the task available in the Methods section.

p=.023). Higher environmental noise reduced learning rates in volatile environments (line with information theoretic predictions [25], F(1,29)=7.496, p=.01) but did not influence learning rates in stable environments (Fig Fi in S1 Text). As expected, there was a main effect of the type of block participants started the task off with influencing this significant main effect of noise (F(1,29)=4.237, p=.049). Participants' self-reported judgements on environmental volatility and noise, collected at the end of each task block, did not reflect the differences between the conditions, and consequently did not reveal any significant main effects or interaction terms (Fig Fii in S1 Text, all p>.280). We reanalysed participant choice behaviour using the exponential-logarithmic model, considering that it demonstrated comparable behaviour with the RW model in the preceding sections, and it can be fully reduced to the RW model as shown in Fig 3 [and legends] making these models overall easier to compare. The model log likelihoods accounting for how well they fit to participant choice behaviour were not significantly different between the exponential-logarithmic model, RW with 4 learning rates (i.e., a separate learning rate for each block) or RW with one learning rate fitted to all blocks all at once (F(2,95)=2.28, p=.108). Other model selection metrics such as Bayesian exceedance probabilities would favour exponential-logarithmic model among all models fitted to all task blocks all at once (xp=0.47), whereas group -wise sum of AIC values favoured models that cannot estimate trial-wise learning rates, i.e., RW model with 1 (AIC: 7667) or 4 learning rates (AIC: 7199) over the exponential logarithmic model (AIC: 7727). However, the fitting of the hybrid Pearce-Hall was worse (AIC: 8087), and an exploratory analysis of the cubic model yielded the worst model fit (AIC: 8856) as it cannot capture the relationship between PEs and learning rates shown in Fig 8A. Bayesian Omnibus Risk was again calculated to be high at 0.9648. Although we did not have enough numbers in each group to statistically compare the models in this domain (i.e., two RW models were globally better fitting), descriptively, the behaviour of the participant who accumulated the highest number of points in the task [14,169 points] was best explained by the exponential logarithmic model, whereas the behaviour of the participant to accumulated the least amount of points in the task [10,803 points] was best explained by the cubic model. These anecdotal results may represent two polar ends of the optimal behaviour spectrum. There were no participants

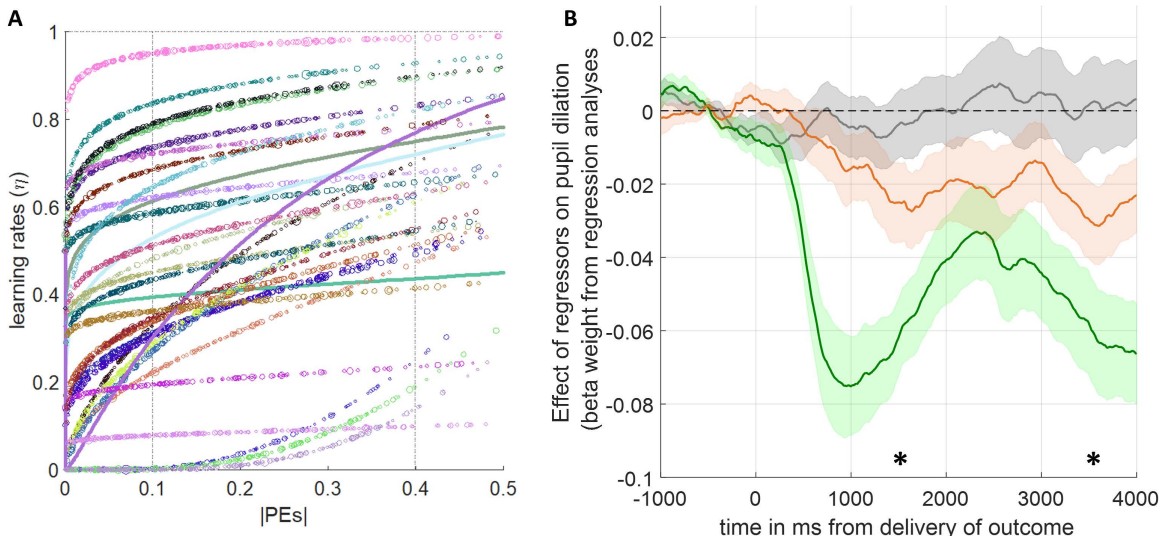

**Fig 8. The learning rates and their pupillary correlates. (A)** Relationship between the absolute value of prediction errors and learning rates. Each single-subject is represented as different colours. Variability in individual marker size along each single-subject trajectory is scaled to the average normalised pupil size during the outcome delivery period at that intersection of prediction errors and learning rates. Four thick continuous lines designate the participants whose pupil data was corrupted, demonstrating only the behavioural relationship. **(B)** Time evolution of regression coefficients during the outcome delivery period: unsigned prediction errors (grey); learning rates (orange); chosen outcome magnitude (green). Error shading designate ±1 SEM. *p<.05 designate bins of 1000ms in which the average pupil dilation is significantly different relative to baseline.

whose behaviour was best explained by the PH model. As a final behavioural control analysis, we further investigated the behaviour of these models in 1000 randomly generated environments in which we manipulated the volatility [via number of switches in the reward mean] and noise [via SD of the generative distribution]. These environments had 60 trials, identical to a single block of our task, also aiming to inform future study designs. The results suggested that volatility and overall reward mean of the environment did not have much influence on which model can track the mean rewards better (both $F_{(2,999)} < 1.02$, both $p > .362$). However, noise was significantly less in environments where models that can generate trial-wise learning rates were able to track the mean rewards (namely exponential-logarithmic and Pearce-Hall; $F_{(2,999)} = 258.32$, $p < .001$). Exponential-logarithmic model performed substantially better than the Pearce-Hall model, i.e., performing better in 196 out of 1000 environments, versus 35 in favour of Pearce-Hall. Simple Rescorla-Wagner model performed the best overall, in 769 out of 1000 environments. Despite Rescorla-Wagner model globally performing substantially better than the exponential-logarithmic model (77% versus 20%, respectively), in real terms the difference between these models equate to a magnitude of 3 pence per trial in prediction errors while tracking the mean the reward in environments where rewards are represented in pence unit, generated by MATLAB's *rand* function. For illustrative purposes, when the reward mean is 50p (±SD) in the environment, the Rescorla Wagner model could estimate it as 49p, whereas the exponential-logarithmic may estimate the mean as 52p (note that models do not systematically under or overestimate the mean reward). The cubic model was not able to win in any of the environments (Fig G in S1 Text).

The relationship between learning rates and prediction errors as estimated by the exponential-logarithmic model is shown in Fig 8A (for each subject in our sample, estimated and plotted individually), and reveals a mixture of trajectories: some resembling those shown for the data from Vaghi et al., in the preceding section, others similar to the parabolic trajectory shown in the re-analysis of data from Behrens et al., and also RW-like constant with a subtle and gradual linear increase over the PEs. This pattern of learning rate/PE relationship was mostly expected as the task design had elements from both of these previously published studies. The learning rate trajectories further highlight that exponential-logarithmic model can flexibly capture individual variability in RL. The PEs generated under this model demonstrates that in environments with variable [as opposed to binary] outcomes, prediction errors encode the structure of the environment (Fig E in S1 Text).

Subsequently, we investigated the physiological correlates and the temporal properties of the learning rates encoded by pupil dilation, along with other key variables such as signed PEs and chosen outcome magnitudes. We used an approach similar to model-based analysis of functional MRI data [41], using an ordinary least squares regression model that we also validated in a number of our previous studies [16,32,42,43]. Our prediction was that learning rates should be instantiated on the same trial (in line with the cubic and exponential-logarithmic models we introduced earlier, and in contrast to the predictions of the Pearce-Hall model), therefore should influence pupil dilation in a gradually increasing manner during the outcome delivery period (i.e., the latter end of the outcome delivery period) while expectations are updated. To capture this effect, we investigated the within-subject correlation between time during the outcome delivery period and regression coefficients for learning rates. As shown in Fig 8B, higher learning rates (orange line) gradually led to pupil constriction during the outcome delivery period (population average for within-subject correlation coefficient $r_{(1998)} = -.2589$ corresponding to $p < 1.57 \times 10^{-27}$; population average r-values significantly lower than zero, $p = .0173$). Average pupil response encoding learning rates was significantly lower than baseline from 1000ms onwards in the outcome delivery period (significance at peak $t_{(27)} = -2.969$, $p = .006$). Here, it is worthwhile to highlight that due to our task design (Fig 7) the chosen outcome magnitudes are represented as bar fillings resulting in greater illumination during the outcome period relative to the fixation cross which marks the beginning of a trial. This is also demonstrated in an even stronger pupil constriction in response to chosen outcome magnitudes (green lines in Fig 8B). Consequently, replication of the pupillary correlates of learning rates in different task designs may yield a positive correlation during the outcome delivery period. Nevertheless, a prediction arising from this work would be that the physiological response to learning rates should continuously build up during the outcome delivery period as previous expectations are updated in the face of new evidence. Signed

PEs encoding the global task structure (Fig E in S1 Text) did not influence pupil dilation (Fig 8B, grey line). Furthermore, we reanalysed the pupillary data with the identical regression model but using the parameter estimates from the hybrid Pearce-Hall model that can also estimate learning rates on a trial-by-trial basis but makes a different temporal prediction about their properties. Learning rates estimated by the Pearce-Hall model did not have a significant physiological correlate (significance at peak t (27) = -1.970, p = .059, Fig H in S1 Text). Therefore, we think pupillary/physiological data lends greater support to the exponential logarithmic model which also had better behavioural model fitting metrics relative to the Pearce-Hall model.

We also report results from iterative stochastic generate-recover simulations which indicated desirable parameter recovery metrics for the exponential-logarithmic model (e.g., stable log likelihood values, Fig Ii in S1 Text) and demonstrated that the model recovery can recapitulate the behavioural pattern observed in the behavioural experiment (Fig Iii in S1 Text).

## Discussion

In this work, we showcase two nonlinear update rules (i.e., cubic or exponential-logarithmic) for human reinforcement-learning (RL). One of these models relied on a strong *a priori* parabolic assumption in describing the relationship between learning rates and prediction errors (PEs), whereas the other was data-driven and displayed even greater flexibility in capturing the individual variability in human learning behaviour. To the best of our knowledge, these models were not previously reported in the literature. Both simulated data (Fig 4), and analysis of human learning behaviour from different datasets [19,40] (in Figs 5–6 and 8) demonstrated that newly proposed models, particularly the exponential-logarithmic model, perform comparably well with the existing RL models while generating reliable trial-wise learning rate estimates from participant choice behaviour. The proposed nonlinear models made a concrete and logical prediction about the temporal properties of learning rates for the epoch immediately after the decision onset (Fig D in S1 Text), and pupillary data we collected during human reward-guided learning lend preliminary evidence to support this prediction (Fig 8B).

We think that these nonlinear update rules will have very important implications for neural sciences. Analysis of various functional neuroimaging modalities (e.g. fMRI, MEG, pupillometry, etc.) heavily relies on general linear models (GLMs) to parse complex higher-order patterns of observed variance in the dependent measure (i.e., the assay of neural activity) into components attributable to a particular source (i.e., experimental or nuisance factors interacting with one another in particular combinations). In modelling functional neuroimaging, GLMs require trial-wise estimates of behavioural processes as predictive variables (i.e., regressors). In GLMs, using decorrelated regressors is vitally important for model efficiency and diminishing collinearity. Consequently, extracting reliable, trial-wise estimates of learning rates that are also decorrelated with PEs is an extremely important issue in neural sciences. We showed that both nonlinear models meet this objective. This feature of the novel nonlinear models is also shared by the hybrid Pearce-Hall model (Eqs 2–3). The learning rates generated under these novel models were all highly significantly correlated with those generated under the hybrid Pearce-Hall model, giving the nonlinear models external validity. Here, it is worthwhile to highlight that we compared these models in environments where reward (*r*) distributions, or naturally the reward probabilities, were in the normalised space (0 < *r* < 1). Future studies may investigate the degree to which these common implementations pose a limitation to the generalisability of these models to real-world scenarios such as consumer behaviour [44].

To the best of our knowledge not many studies independently manipulated the sources of expected and unexpected uncertainties in learning environments (such as the task environment shown in Figs 4 and S5 in S1 Text), without explicitly informing participants about the properties of the task environment [45,46] in which they were asked to perform reinforcement-learning. Future studies using our 2x2 (volatility and noise) full factorial approach can probe where different components of RL models are encoded in the human brain more precisely (i.e., striatal regions, habenula [encoding negative PEs] [47], bilateral amygdala and dACC), and where this information is integrated in the brain. We think that our more flexible and data-driven exponential-logarithmic model can lead the way, as it can be fully reduced to the RW

model (Fig 3) if the relationship between PEs and learning rates happen to be constant and linear when estimated from participant choice behaviour. Consequently, we think that benefits of relying on a 2-parameter exponential-logarithmic model to generate reliable trial-wise learning rate estimates markedly outweigh the lower complexity afforded by the traditional 1-parameter RW model. Although we provided some preliminary evidence to support temporal predictions based on this model (Fig 8B), at this moment in time we were not able to include functional neuroimaging to be able to demonstrate the full neural circuitry associated with it [due to funding limitations]. However, we think that the neuroimaging directions that we outline here are important not only for cognitive/computational neuroscience but also for promoting a better understanding of psychiatric disorders in which learning processes are impaired. Furthermore, it is important to highlight that, analytically, the pupillary data is similar to fMRI data obtained from a single voxel, so future neuroimaging studies using these models can reveal important insights about a wider neural circuitry associated with RL and integration of value information during learning.

It might be worthwhile to discuss the merits of the cubic model, separately from the commentary above. This elegantly expressed model (Eq 5) may be particularly useful in tasks in which only the expected uncertainty (i.e., noise) is experimentally manipulated [45,46] with a finer gradient in magnitude learning tasks, while keeping the average reward amount associated with one shape stable at different levels (e.g., 0.1, 0.2, 0.3, 0.4 …). In our own lab-based experiment (Figs 7 and S6 in S1 Text), we did not focus on systematically covering the average reward x noise (i.e., SD) space and our choice of 0.5 as average reward rate in stable environments inevitably pushed towards maximum entropy. Experiments systematically covering this stimuli space should be considered for future research. In such experiments, the parameter $\sigma$ (Eq. 2 and Fig 2) can be estimated from participant choice behaviour as the single model parameter which accounts for the participants' inference about the level of noise/expected uncertainty in the task environment. This would help us to understand how participants own understanding of environmental noise as assessed by subsequent self-reports and as parameterised in a learning model map on to actual noise level in the environment. In this preliminary work, we showed that participants did not have a good, higher-order understanding of environmental volatility and noise, despite showing near-optimal learning behaviour (Fig Fii in S1 Text). The output of our cubic model (i.e., variable, trial-wise learning rates) is similar to a recently described neurophysiological reward-dependent metaplasticity model that can account for the learning behaviour of non-human primates [48]. However, under the cubic model the level of expected uncertainty in the environment determines the *transition gradient* with which agents should attribute higher PEs to unexpected uncertainty/ volatility (Fig 2), instead of expected uncertainty setting a binary *detection baseline* for changes which should be attributed to environmental change [24]. Needless to highlight, the cubic model is still much easier implement on data collected from human volunteers due to feasibility limitations in electrophysiological recording. Overall, although our recommendation for learning scholars would be to rely on the exponential-logarithmic model, the cubic learning rule can also be a viable alternative as it is simple, intuitive and can extract trial-wise learning rates and PEs which are decorrelated with each other. Particularly, in RL tasks in which participants are expected to learn hidden probabilities such as the one used by Behrens and colleagues, the cubic model's a priori assumptions may be reasonably accurate as we showed in Fig 5B.

In the past two and a half decades, cognitive computational neuroscience has made big advances by studying the neural correlates of PEs. Consequently, the number of studies that investigated brain regions encoding PEs significantly outnumber those that investigated neural regions encoding learning rates [49]. One study using the hybrid Pearce-Hall model described in this paper in a reversal-learning task involving aversive outcomes (a stable 30% probability of receiving an electroshock) showed that learning rates are encoded in the human amygdala, bilaterally [13]. On the other hand, three key studies that manipulated the volatility (i.e., the unexpected uncertainty) of reward learning environments showed that volatility of the environment, which in return correlates significantly with learning rates, are encoded in the dorsal and subgenual anterior cingulate cortex (ACC) in both non-social and social learning environments [5,19,49]. Recently, we have reviewed learning and value-based decision-making literature across rodent, monkey and human imaging studies and our synthesis also pointed out to a role for dACC potentially guiding learning rates [47]. We hope that models that we

communicate in this manuscript will ignite further interest in exploring the neural correlates of learning rates as opposed to PEs, and in doing so, we will help to define the next decades of functional neuroimaging and electrophysiology research.

Finally, we also hope these models can aid understanding behaviour in psychiatric conditions associated with learning impairments. A recent computational meta-analysis indicated that RL impairments in patients with depression and anxiety load on to reward sensitivity rather than learning rates [50]. However, our current understanding of impairments in psychiatric conditions rely on the assumption that learning rates are constant, as the wide majority of computational psychiatry work relies on learning rates estimated by the RW model. Here, we showed different lines of evidence that this may not be correct and utilising our models to analyse the relationship between learning rates and PEs can pave the way to novel insights in mental health conditions that remain currently unknown.

## Methods

### Ethics statement

The following experimental procedures were approved by the institutional review board of University of Oxford (Central University Research Ethics Committee, R49753/RE001). Prior to the experiment, participants were given the Participant Information Sheet (PIS) clearly describing the details of the study. After reading the PIS and having an opportunity to ask any related questions, all participants signed an informed consent document.

### Reanalysis of existing datasets

In the reanalyses of behavioural data from Behrens et al., (2007) a maximum likelihood approach was used, which was implemented by running MATLAB's *fmincon* function iteratively with random starts covering the whole parameter space (1000+iterations per participant per condition). No trials or participants were excluded from the analysis. In the reanalysis of behavioral data from Vaghi et al., (2017) parameters were estimated in a similar manner to maximum likelihood approach (i.e., identifying the parameter combination that minimises the delta between model estimates and participants' actual bucket positions in the normalised space).

### Pupillometry experiment

The study involved a single experimental session during which participants completed a novel reward learning task. The task was displayed on a VGA monitor. Participants' heads were stabilized during the task while an eye-tracking system (Eyelink 1000 Plus; SR Research, Ottawa, Canada) was recording pupillometry. The pupil dilation was recorded at a sampling rate of 500 Hz. The VGA monitor was linked to the laptop running the experiment with Psychtoolbox on MATLAB.

The task consisted of 4 blocks of 60 trials each. The decision time was self-paced, with additional rest period after every block. Participants were presented with two abstract shapes from the Agathodaimon font (Fig 7). The abstract shapes of the learning task were offset by around 7 visual angle and were displayed on either side of an empty black score bar as well as a fixation cross indicating the middle of the screen during the inter-trial interval. The shapes were presented randomly on each side of the screen, prompting participants to identify the better shape rather than right versus left side of the screen. The outcome reward magnitude following participant choice was displayed inside the empty bar between the options, as a proportional green filling for a jittered interval of 2–6 seconds. The shapes were the same for each trial during a given block but changed into the new block (total of 2x4 shapes). At the end of each trial, participants' running total was updated on the screen. The order in which all the blocks were completed was counterbalanced across participants. The information content of the reward outcomes was varied across the four blocks by altering the volatility and noise associated with the stimulus-outcome pair. The outcomes for each block were thus generated following a 2 x 2 factorial of high/low volatility and high/low noise schedule (Fig E in S1 Text). For the entire task, the quantity and variety of stimuli remained constant in order to minimise disparities in luminance between trials. At the end of the experiment, the

outcome of 20 randomly selected trials were converted to real money which was added to a baseline payment of £10. The participants were told to bank as many points as possible at the beginning of the experiment. After each block, participants were asked to rate the number of times they thought the average reward shifted from low to high, and how wide they thought the range of rewards were around the average reward value (Fig Fii in S1 Text).

### Pupillometry preprocessing and analysis

We followed identical steps to our previous eye tracking studies in preprocessing and analysing pupillary data [16,32,42,43]. Pupillometry data was collected to examine the participants' physiological responses during the reward-guided learning task. Eyeblinks were removed from the data by using the Eyelink system's built-in filter. This was followed by a linear interpolation to compensate for missing data points. The resulting pupillary trace was processed through a low pass Butterworth filter with a cut-off of 3.75 Hz, and then z-transformed across the experimental session [51]. The pupillary correlates of the model-based regressors were assessed by fitting a multiple linear regression model on baseline corrected pupil data. If over half of the data from the outcome period had been interpolated, the affected trial was excluded from the analysis. Participants who had missing data more than 50% of the trials were excluded from the analysis, leading to the exclusion of four cases in total.

The pupillary multiple linear regression model had the following regressors: an intercept, trial number (as a proxy for tiredness), chosen reward outcome, PEs and learning rates and average number of black pixels displayed on otherwise grey/neutral screen as shown in the task timeline (Fig 7).

## Supporting information

**S1 Text. Table A. Demographic details of participants. Fig A.** The relationship between environmental noise and parameter κ. **Fig B.** The average magnitude of prediction errors under different learning models. **Fig C.** A nonlinear relationship between the absolute magnitude of prediction errors and learning rates. **Fig D.** Learning rate acceleration differences between hybrid Pearce-Hall, cubic and exponential-logarithmic models. **Fig E.** Prediction errors encode the structure of the environment. **Fig F.** Behavioural results from the in-lab experiment. **Fig G.** The results of simulations highlighting the descriptive features of environments which favour one model over the others. **Fig H.** Supplementary control analysis of the pupil dilation with the outputs of the Pearce-Hall model. **Fig I.** Outputs of the generate-recover simulations. (PDF)

## Acknowledgments

The authors would like to thank to Prof. Tim Behrens and Dr Matilde Vaghi for sharing the behavioural data and a helpful discussion. The authors would also like to thank to Prof. Peter Dayan, Prof Michael Browning and the participants of the Oxford Computational Psychiatry Journal Club for their helpful comments about behavioural modelling.

## Author contributions

**Conceptualization:** Erdem Pulcu.

**Formal analysis:** Boluwatife Ikwunne, Erdem Pulcu.

**Funding acquisition:** Boluwatife Ikwunne, Erdem Pulcu.

**Investigation:** Boluwatife Ikwunne, Jolie Parham, Erdem Pulcu.

**Methodology:** Erdem Pulcu.

**Software:** Erdem Pulcu.

**Supervision:** Erdem Pulcu.

**Visualization:** Erdem Pulcu.

**Writing – original draft:** Erdem Pulcu.

**Writing – review & editing:** Boluwatife Ikwunne, Jolie Parham.

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
