## [Decision Letter · Decision Letter 0]

19 Feb 2025

PCOMPBIOL-D-24-01275

A nonlinear relationship between prediction errors and learning rates in human reinforcement-learning

PLOS Computational Biology

Dear Dr. Pulcu,

Thank you for submitting your manuscript to PLOS Computational Biology. After careful consideration, we feel that it has merit but does not fully meet PLOS Computational Biology's publication criteria as it currently stands. Therefore, we invite you to submit a revised version of the manuscript that addresses the points raised during the review process.

Please submit your revised manuscript within 60 days Apr 21 2025 11:59PM. If you will need more time than this to complete your revisions, please reply to this message or contact the journal office at ploscompbiol@plos.org. Please include the following items when submitting your revised manuscript:

We look forward to receiving your revised manuscript.

Kind regards,

Camilla L Nord

Guest Editor

PLOS Computational Biology

Marieke van Vugt

Section Editor

PLOS Computational Biology

**Additional Editor Comments :**

I would first like to apologise on behalf of the journal for the amount of time this has taken, which was due to PLOS CB being required to find a new section Editor. I agreed to Guest Edit this manuscript in January; in this capacity, I returned it to its original reviewers following your extensive revisions.

I have read each Reviewer comment in detail. Reviewer 1 has only minor remaining concerns, and believes the manuscript is now stronger. Reviewers 2 and 3 still have substantial concerns which would need to be addressed before considering publication. For this reason, I would like to give you an option to revise the manuscript, if you believe their comments can be addressed. Of course, I would also understand if you felt that the piece would be a better fit elsewhere given this. In the context of a resubmission, the two biggest concerns as I see it are:

(1) Reviewer 2 and Reviewer 3’s shared concern about whether or not these models outperform existing ones in any context. Although further simulations might help address this, an alternative/additional way to address this comment is Reviewer 2’s suggestion of reframing the manuscript. Reviewer 2 suggests that, even if reframed substantially in this way, the manuscript would still be a novel contribution to the literature.

(2) Reviewer 3’s comment about ‘where the model (and its assumptions) come from’ – I think this is an interesting point and could be addressed in a rewrite including, for example, complementary neurobiological evidence or theories.

Other areas mentioned seem addressable to me, such as additional details for the simulation (Reviewer 3), editing the introduction and some captions to make them more concise and possibly in a better order (as suggested by Reviewers 2 and 3), potentially rewriting some of the language (due to the stylistic concerns of Reviewer 3), and further justification/simplification of the model comparison process (Reviewer 2). Please also ensure the OSF link contains the most relevant/up to date materials.

**Journal Requirements:**

At this stage, the following Authors/Authors require contributions: Erdem Pulcu, Boluwatife Ikwunne, and Jolie Parham. Please ensure that the full contributions of each author are acknowledged in the "Add/Edit/Remove Authors" section of our submission form.

4) We notice that your supplementary Figures, and Table are included in the manuscript file. Please remove them and upload them with the file type 'Supporting Information'. Please ensure that each Supporting Information file has a legend listed in the manuscript after the references list.

1) State what role the funders took in the study. If the funders had no role in your study, please state: "The funders had no role in study design, data collection and analysis, decision to publish, or preparation of the manuscript.".

6) Your current Financial Disclosure states, " EP is full-time employed on a grant jointly funded by the UK Medical Research Council and Janssen Pharmaceuticals (MR/S035591/1) awarded to Catherine J Harmer, Susannah Murphy and Philip J Cowen. BI is supported by Rhodes Scholarship for West Africa. The pupillometry study was funded by a University of Oxford Department of Psychiatry small research grant awarded to BI and EP." However, your funding information on the submission form indicates receiving one grant.

Please ensure that the funders and grant numbers match between the Financial Disclosure field and the Funding Information tab in your submission form. Note that the funders must be provided in the same order in both places as well.

**Reviewers' comments:**

Reviewer's Responses to Questions

**Comments to the Authors:**

**Please note that one of the reviews is uploaded as an attachment.**

Reviewer #1: The authors have done a good job addressing most of my comments, and I believe that the paper is now stronger. I have a few remaining minor points:

- Regarding model comparisons, I appreciate that the authors have more consistently compared all models on all of the datasets, and I believe that this allows for much stronger conclusions to be drawn. Yet the cubic model is not compared on the third dataset, despite it being the only model that demonstrates higher performance than all other models on at least one metric (on the Behrens data, via AIC the cubic model is slightly better than the 'runner-up' pearce-hall model and at least somewhat better than the other models, although these values are not reported; on the Vaghi data, of their models the exp-log is best but it performs equally to rescorla-wagner). Given this, I don't think it is fully justified to exclude this model from the comparisons on the final dataset, so I believe it should be included here, both for completeness/consistency but also because the cubic model outperformed the exp-log model on the Behrens data (when fit continuously across blocks).

- On line 398: It is stated that cubic model has three free parameters, including the learning rate, but isn't the learning rate determined by equation 4, which depends on the free parameter kappa, making it not a free parameter?

- The cubic model is still referred to as the "Parabolic" model throughout the figures--this should be changed to match the text in order to be more clear.

Reviewer #2: The review is uploaded as an attachment.

Reviewer #3: Disclaimer:

This paper seems to be a new submission which was submitted before to the same journal (noted as a new manuscript ID). I was one of the original reviewers (Review #1) for ther early submission.

The authors did try to address as many review comments as possible. But in my view, this manuscript, in its current form, is still far from being ready to be published.

At large, the writing style does not seem to be motivated as a scientific publication. In many places, the authors use phrases like “We think […]” “Let us start by …”, which, with due respect, are very much written as an essay rather than a academic report. If the authors were choosing to rewrite it, the manuscript will appear to be substantially different than the current one, which may again trigger the action as a new submission rather than a revision. I leave this for the editor to decide which is the best way to proceed.

Relatedly, there are multiple jumps in the flow of the writing. For example, the key concept of learning rate is mentioned before being properly explained/introduced. On line 83, it is the first time the term learning rate is mentioned, but only until line 93-94, learning rate is given a definition. Also, I would not use the acronym LR for learning rate, as, in the manuscript, there are both RL (reinforcement learning, which is a way more common acronym) and LR (learning rate), which is rather confusing to read.

A rather critical issue is where the main model comes from. It seems as if one came up with a model out of no where, then it happened to work. Although there are indeed some assumptions (starting from line 176), there exists multiple non-linear functions that may also be plausible. Are there any theories, neuro/biological evidence to suggest the proposed model is the best choice?

I also noticed that the authors have toned down that the proposed model is not superior to existing one, but rather a alternative. This somewhat undermines the value of this work, and impacts the potential to be eventually published in this journal.

Moreover, some details are still lacking for the simulation. It seems that an algorithm (start from line 298) is used to minimize PE, without using functions such as softmax. But then how are actions/choices simulated? Some form of choice rule must have been applied? OR the model only cares about “value” rather than action?

Very last (sort of minor), the statement of “PEs are always between -1 and 1” (line 178 ) is incorrect. The range of PE is dependent on how outcome/reward is coded. If, as one example, in a task/model the reward ranges from 0 to 5, then the range of PE is -5 and 5.

**Have the authors made all data and (if applicable) computational code underlying the findings in their manuscript fully available?**

Reviewer #1: Yes

Reviewer #2: **No: ** The link in the paper leads to an OSF repo which has code from 2019 and does not appear to include any data.

Reviewer #3: Yes

PLOS authors have the option to publish the peer review history of their article (what does this mean? ). If published, this will include your full peer review and any attached files.

**Do you want your identity to be public for this peer review?** For information about this choice, including consent withdrawal, please see our Privacy Policy .

Reviewer #1: No

Reviewer #2: No

Reviewer #3: No

**Figure resubmission:**
---

## [Decision Letter · Decision Letter 1]

20 Jun 2025

PCOMPBIOL-D-24-01275R1

A nonlinear relationship between prediction errors and learning rates in human reinforcement-learning

PLOS Computational Biology

Dear Dr. Pulcu,

Thank you for submitting your manuscript to PLOS Computational Biology. After careful consideration, we feel that it has merit but does not fully meet PLOS Computational Biology's publication criteria as it currently stands. Therefore, we invite you to submit a revised version of the manuscript that addresses the points raised during the review process.

Please submit your revised manuscript within 30 days Aug 20 2025 11:59PM. If you will need more time than this to complete your revisions, please reply to this message or contact the journal office at ploscompbiol@plos.org. Please include the following items when submitting your revised manuscript:

We look forward to receiving your revised manuscript.

Kind regards,

Camilla L Nord

Guest Editor

PLOS Computational Biology

Marieke van Vugt

Section Editor

PLOS Computational Biology

**Additional Editor Comments:**

Many thanks for your resubmission. From your previous three reviewers, one was satisfied with the manuscript as is, so I re-invited the two with remaining concerns to read your resubmission. Of those two, you will see that your changes have addressed many issues, but both do have a smaller number of outstanding issues many of which I still think might improve the manuscript if addressed, including regarding OSF code and perhaps most importantly clarifying the overall motivation. However, I hope resolving these issues for the most part would be relatively minor, and that following these being addressed, the paper will be suitable for publication in PLOS Computational Biology. Please let me know if you do not feel you are able to address these; otherwise, I will expect a response outlining what was changed (with the specific changes under each point, please) for me to make a final decision. Thank you.

**Journal Requirements:**

- Please ensure that the funders and grant numbers match between the Financial Disclosure field and the Funding Information tab in your submission form. Note that the funders must be provided in the same order in both places as well.

2) State what role the funders took in the study. If the funders had no role in your study, please state: "The funders had no role in study design, data collection and analysis, decision to publish, or preparation of the manuscript.".

**Reviewers' comments:**

Reviewer's Responses to Questions

**Comments to the Authors:**

Reviewer #2: I thank the authors for their clarifying comments. These have not, however, changed my fundamental view of the issues. As such, I am still unconvinced that the manuscript is ready for publication; however, given that the disagreements are well spelled out at this point, I do not believe a further round of reviews would necessarily be helpful and would leave it up to the editors to decide whether to go forward. Nonetheless, I provide a few more comments (including some reiterations of my previous comments) in the hope that the authors will find them useful.

Major

The motivation remains unclear to me. I’m not trying to be facetious, but I genuinely struggle to understand the paper's main aim – whether it aims to describe human learning accurately (which the authors suggest Bayesian models do better) or to provide more usable, intuitive models. If the latter, then conceptual/psychological interpretability of all free parameters is essential, as is superior model fit and model validation because they are necessary to be reasonably confident that the variable learning rates the models provide are meaningful. This is particularly important if the goal is to track individual differences in the relationship between learning rates and prediction errors – which their exponential-logarithmic model enables (which is great!) –, or to track learning rate signatures in the brain. At a minimum, the authors should demonstrate that the RW and exponential-logarithmic models are distinguishable in model recovery analyses using realistic parameter ranges (drawn e.g., from their simulations and reanalyses).

Minor

• There is no real justification for the non-traditional fitting approach (why not use generate-recover allowing traditional likelihood estimation?) and for minimizing prediction errors across the entire trajectory when this differs from experimental data analysis. As an experimentalist, this puzzles me. I would like to understand when this model would be preferable to other models, so simulations should examine simulated experimental data with plots showing both simulated and fitted behavior

• It seems improper to cluster models by parameter count for model comparisons (lines 413ff). I would still urge the authors to consistently use model comparison that penalize complexity, including parameter number

• I maintain that much information provided in the figure legends belongs in the main text (or it risks being overlooked)

• Figure 3 could really use a legend identifying the delta and lambda combinations for at least 3-5 lines. I may be dense, but the psychological meanings of lambda and delta remain unclear to me (unlike the parabolic and cubic model parameters, which are well described)

• It would be helpful if Figure 6 could show trial-by-trial learning rates for both RW (flat) and exponential-logarithmic models to illustrate their differences, plus within-subject standard deviations of learning rates. Since the exponential-logarithmic model can reduce to RW under certain conditions, these comparisons would show how substantially they really differ.

• Please have another sweep for typos, there were a few stray ones.

• Finally, it would be really helpful if the authors highlighted the changes made in response to the reviewer comments in the response letter – this would save reviewers a lot of time.

Reviewer #3: Although I do not fully agree with the style of writing, which is in essence the authors' choice, and having paper published in such a style previously does not mean this is the preferred way to communicate science, I do not object this MS being published.

As the authors have claimed, they believe this model/approach can be very useful for the field.

However, the current open code on OSF is rather disorganized - neither a Readme file nor enough documentation was provided. I would strongly encourage the authors to tidy up the code and write clear documentations of their matlab scripts, before the manuscript can be eventually published. This way, I believe the field can benefit from it at large.

**Have the authors made all data and (if applicable) computational code underlying the findings in their manuscript fully available?**

Reviewer #2: Yes

Reviewer #3: Yes

PLOS authors have the option to publish the peer review history of their article (what does this mean? ). If published, this will include your full peer review and any attached files.

**Do you want your identity to be public for this peer review?** For information about this choice, including consent withdrawal, please see our Privacy Policy .

Reviewer #2: No

Reviewer #3: No

**Figure resubmission:**
---

## [Editor Report · Decision Letter 2]

19 Aug 2025

Dear Dr Pulcu,

We are pleased to inform you that your manuscript 'A nonlinear relationship between prediction errors and learning rates in human reinforcement-learning' has been provisionally accepted for publication in PLOS Computational Biology.

Best regards,

Camilla L Nord

Guest Editor

PLOS Computational Biology

Marieke van Vugt

Section Editor

PLOS Computational Biology

Many thanks for your thorough revisions to the manuscript, and congratulations on your excellent work.

---

## [Editor Report · Acceptance letter]

PCOMPBIOL-D-24-01275R2

A nonlinear relationship between prediction errors and learning rates in human reinforcement-learning

Dear Dr Pulcu,

I am pleased to inform you that your manuscript has been formally accepted for publication in PLOS Computational Biology. Your manuscript is now with our production department and you will be notified of the publication date in due course.

With kind regards,

Anita Estes
